# In Vitro Fertilisation (IVF) Associated with Preimplantation Genetic Testing for Monogenic Diseases (PGT-M) in a Romanian Carrier Couple for Congenital Disorder of Glycosylation Type Ia (CDG-Ia): A Case Report

**DOI:** 10.3390/genes11060697

**Published:** 2020-06-25

**Authors:** Bogdan Doroftei, Loredana Nemtanu, Ovidiu-Dumitru Ilie, Gabriela Simionescu, Iuliu Ivanov, Emil Anton, Maria Puiu, Radu Maftei

**Affiliations:** 1Origyn Fertility Center, Palace Street, no 3C, 70032 Iasi, Romania; bogdandoroftei@gmail.com (B.D.); nemtanuvalentinaloredana@gmail.com (L.N.); gabi.ginecologie@gmail.com (G.S.); iuliuic@gmail.com (I.I.); emil.anton@yahoo.com (E.A.); dr.radu.maftei@gmail.com (R.M.); 2Department of Mother and Child Medicine, Faculty of Medicine, University of Medicine and Pharmacy “Grigore T. Popa”, University Street, no 16, 700115 Iasi, Romania; 3Clinical Hospital of Obstetrics and Gynecology “Cuza Voda”, Cuza Voda Street, no 34, 700038 Iasi, Romania; 4Department of Molecular Genetics, Faculty of Biology, University of “Alexandru Ioan Cuza” Carol I Avenue, 700505 Iasi, Romania; 5Department of Research, Faculty of Biology, Alexandru Ioan Cuza University, Carol I Avenue, no 11, 700505 Iasi, Romania; 6Regional Oncology Institute Iasi, Department of Molecular Biology, General Henri Mathias Berthelot Street, no 2-4, 700483 Iasi, Romania; 7Department of Microscopic Morphology, Faculty of Medicine, University of Medicine and Pharmacy “Victor Babeș”, Eftimie Murgu Square, no 2, 300041 Timișoara, Romania; maria.puiu@gmail.com; 8Department of Morphofunctional Sciences, Faculty of Medicine, University of Medicine and Pharmacy “Grigore T. Popa”, University Street, no 16, 700115 Iasi, Romania

**Keywords:** in vitro fertilisation, preimplantation genetic testing for monogenic diseases, congenital disorder of glycosylation type Ia

## Abstract

**Background:** Congenital disorder of glycosylation (CDG) is a severe morphogenic and metabolic disorder that affects all of the systems of organs and is caused by a mutation of the gene *PMM2*, having a mortality rate of 20% during the first months of life. **Results:** Here we report the outcome of an in vitro fertilisation (IVF) cycle associated with preimplantation genetic testing for monogenic diseases (PGT-M) in a Romanian carrier couple for CDG type Ia with distinct mutations of the PMM2 gene. The embryonic biopsy was performed on day five of the blastocyst stage for six embryos. The amplification of the whole genome had been realized by using the PicoPLEX WGA kit. Using the Array Comparative Genomic Hybridisation technique, we detected both euploid and aneuploid embryos. The identification of the PMM2 mutation on exon 5 and exon 6 was performed for the euploid embryos through Sanger Sequencing with specific primers on ABI 3500. Of the six embryos tested, only three were euploid. One had compound heterozygosity and the remaining two were simple heterozygotes. **Conclusion:** PGT-M should be strongly considered for optimising embryo selection in partners with single-gene mutations in order to prevent transmission to the offspring.

## 1. Introduction

Congenital disorder of glycosylation (CDG) represents a multisystemic group of mild to severe neuro(metabolic) disorders, inherited in an autosomal recessive manner [1]. Forty-six relevant genes being so far identified [2]. Since the first case reported by the paediatrician Jaak Jaeken [3], remarkable discoveries have been made during these three and a half decades regarding CDG’s underlying mechanism [4].

Mechanically speaking, CDG is defined by a series of defects occurring in the synthesis of the glycan moiety of the glycoconjugates and the coupling of these polysaccharides to proteins and lipids [5]. The latest studies have revealed that CDGs are divided into O-linked disorders, as a result of an attachment of the O-glycans to the hydroxyl group of threonine or the serine of proteins, a process that takes place within only one cellular compartment. It is tissue-specific and possesses a distinct clinical panel, less severe than that of the N-linked group [6].

However, it took fifteen years to unveil the gene responsible for PMM2-CDG [7] (OMIN # 212065), and it has been established that PMM2, which is located on chromosome 16 [8], is the result of an attachment of N-glycans to an amino group of asparagine of proteins. Compared with O-linked, which takes place only in the Golgi apparatus, the N-linked occurs within the cytosol, endoplasmic reticulum, and Golgi apparatus.

More precisely, PMM2 encodes phosphomannomtase (PMM) during the second step of the mannose pathway [9] by converting man-6-P into the Man-1-P necessary to produce guanosine diphosphate (GDP)-Man for the mannosylation of glycans [8].

Among all twenty-five pure N-linked disorders that have been highlighted [5], the PMM2 phenotype is the most frequently encountered [6], with almost 100 mutations known to date [10]. More than 800 cases have been reported worldwide [11], with a frequency rate of 1:12,000, and up to 1:20,000 births [12].

As previously mentioned, several cohort studies have already been conducted in different countries in which both related and unrelated individuals were used [13,14,15,16,17,18,19,20,21,22,23]. Given that most of the reports have focused on the paediatric population [24,25,26], the long-term course during adolescence, respectively, adulthood has also provided additional conclusive data [13,19,22,25,27,28,29,30].

Cumulatively, this study aims to report the case of a Romanian carrier couple for CDG-Ia who underwent an IVF cycle associated with PGT-M by highlighting the necessity for genetic counselling in the field of rare diseases in order to develop adequate services.

## 2. Case Presentation

A 28-year-old female and her 30-year-old partner were presented within Origyn Fertility Center with a recommendation for genetic counselling—in vitro fertilization (IVF) combined with preimplantation genetic testing for monogenic diseases (PGT-M) and, respectively, the identification of the c.470T>C and c.385G>A in embryos. The initial consultation aimed to assess the couple’s medical history, possible risk factors, as well as the results of the previous investigations.

According to their medical history, the couple previously achieved two spontaneous pregnancies, but, unfortunately, both foetuses died at two and six months after birth. The echographic examination during the first semester was normal, with no visible abnormalities in terms of the length of the long bones and cranial biometry. However, during the second trimester, analogous for both pregnancies, discrepancies were observed regarding the length of the long bones and cranial biometry. Even though the first pregnancy did not require hospitalisation, an admission was necessary at 17 weeks during the second one.

Moreover, with the suspicion of chromosomal abnormalities existing, an amniocentesis and prenatal screening were recommended. No molecular aneuploidies of chromosomes 13, 18, and 21 were detected.

The third-trimester examination revealed a pronounced discrepancy between the length of the long bones, cranial and abdominal biometrics, and non-immune hydrops fetalis (HF) diagnosis. In the case of the second foetus, a massive polyhydramnios was observed.

Considering the medical history and death of the first foetus, a series of extensive genetic investigations were carried out for the second one because of the suspicion of metabolic disease. Although there were suspicions of Gaucher disease, the genetic tests conducted ultimately confirmed that the foetus suffered from the CDG-type 1 syndrome.

The necropsy exam revealed cerebral edema, venous stasis in the kidneys, lungs, and liver. The histopathological examination confirmed the presence of hyperemia in the cerebellum, brain, liver, and lung. The second foetus also displayed foetal abnormalities, such as the persistence of the arterial canal and ventricular septal defect.

The genetic investigations carried out for the couple have revealed that both of them are carriers for the congenital disorder of glycosylation, with different mutations of the PMM2 gene, more precisely on exon 5 (father) and exon 6 (mother).

In light of the information obtained from the anamnesis, the couple was counselled by an interdisciplinary team in order to undergo an in vitro fertilisation cycle, followed by an embryo biopsy and preimplantation genetic testing for monogenic diseases (PGT-M).

The partners were admitted to a viral screening for hepatitis and HIV (HBs Ag, Anti-HBc, Anti HCV, Anti HIV1 + 2, VDRL), both testing negative. Further on, the paraclinical investigations involved the determination of endocrine markers (TSH, progesterone, estradiol, FSH, LH, prolactin), the values ranging between normal limits. The ovarian reserve had been previously assessed by determining the anti-Müllerian hormone (AMH), the result being 0.66 ng/mL. The TORCH screening (Rubeola IgG, Varicella IgG, Toxoplasma IgM and IgG, Cytomegalovirus IgG, Herpes IgG) revealed that the male partner was positive for Cytomegalovirus (IgG 10.2 UA/mL) and Rubella (IgG anti-Rubella 97.1 UI/mL), and negative for Herpes, Toxoplasma, and Listeria.

Additionally, the haematological investigation (hemoleucogram, APTT, fibrinogenemia, 25-OH-vitamin D, coagulogram composed of International Normalized Ratio, Quick time) were performed and the results were in normal limits, according to sex and age. The semen quality was previously evaluated, the results indicating normozoospermia.

According to the internal ethic protocol, the medical team offered comprehensive information regarding the intracytoplasmic sperm injection technique, the possible risks related to ovarian stimulation, the chances of obtaining a pregnancy, as well as the possible risks associated with the embryo biopsy and of PGT. The couples signed an informed consent, according to which a single embryo would be transferred if no genetic abnormalities were detected by PGT-M. In the case of a positive result after PGT-M, the procedure would be ceased.

The treatment was carried out by using a GnRH antagonist protocol. The ovarian stimulation was started on the second day of menstruation, with a daily dose of 200 UI rFSH (Puregon, Merck Sharp & Dohme, NJ, USA) and 150 UI rFSH (Menopur, Merck Sharp & Dohme, NJ, USA) until day 10. The final oocyte maturation was triggered following the administration of 10.000 hCG (Pregnyl, Merck Sharp & Dohme, NJ, USA).

Ultrasound monitoring of the ovarian stimulation revealed a total of 17 follicles, with 14 oocytes being retrieved 36 h after the trigger had been administered. Subsequently, the oocytes were inseminated using Intracytoplasmic Sperm Injection (ICSI) and the resulting embryos were cultured and monitored by a time-lapse system (EmbryoScope, Vitrolife, FertiliTech, Arhus, Denmark) until day 5. A total of 11 oocytes had normal fertilisation, and by day 5, four embryos degenerated. Therefore, seven embryos have been biopsied, three of them on day 5 and the other four on day 6 at blastocysts stage.

Agilent oligonucleotide arrays were used according to the instruction of Agilent GenetiSure Pre-Implantation Array-Based CGH for Aneuploidy Screening (Agilent Technologies, Santa Clara, CA. USA). The DNA amplification step uses a PCR-based method optimised for samples containing 3–10 human cells collected from a day 5 trophectoderm. Two experimental samples were hybridised on the same array and compared to a male and female reference sample co-hybridised to another array on the slide, for a total of 14 experimental samples on one 8 × 60 K slide. The PicoPLEX WGA kit (PicoPLEX WGA Kit; Rubicon Genomics, Ann Arbor, MI, USA), which increases the amount of DNA, was used for the whole genome amplification, as well as for the DNA samples of the female and male reference. The amplified DNA was labelled by random priming using either Cy5-dUTP or Cy3-dUTP. After columns-purification, probes were denatured and pre-annealed with human Cot-1 DNA. Hybridisation was performed using an 8 × 60 K slide at 67 °C for 16 h. The array was washed and scanned using the Agilent DNA micro-array scanner and analysed with Feature Extraction Software (Agilent Feature Extraction Software (v10.7), Santa Clara, CA, USA). Subsequently, the results were interpreted with DNA analytics Software.

Of the seven embryos analysed by PGT-M, only three embryos were euploids (Figure A1, Figure A2, Figure A6), one of which presented compound heterozygosity (PMM2 exon 5 and PMM2 exon 6) (Figure A7), while the other three were simple heterozygotes, one PMM2 exon 5, and the other PMM2 exon 6 (Figure A3, Figure A4, Figure A5). After an interdisciplinary consultation, it was decided to transfer the embryo—the carrier of the mutation on exon 5, c.385G>A.

Prior to the transfer of the chosen embryo, hormone replacement therapy was used. On the first day of the menstrual cycle, the endometrial preparation was initiated by using oral estradiol valerate (Cyclo-Progynova; Bayer Pharma AG, Berlin, Germany) 2 mg/t.i.d and, on day 7, the endometrial thickness was evaluated through a transvaginal ultrasound. Accordingly, 2 mg/q.i.d was prescribed until day 15, and beginning with day 16, natural micronized progesterone (Utrogestan; Laboratories Besins International, Paris, France) was started at a dose of 600 mg/day. The chosen embryo (W8) was thawed, evaluated, and transferred on the fifth day of progesterone administration. Fifteen days after the embryo transfer, a biochemical pregnancy was confirmed by determining the β-hCG, 55.60 mUI/mL. Subsequently, a second β-hCG was carried out 48 h later, the results being 36.75, which indicated that the pregnancy was not developing properly. Eventually, the pregnancy was clinically unconfirmed by ultrasound echography. The patients have discussed the outcome of the procedure with the medical team and it was determined that, in the future, the transfer of the remaining euploid embryos may be attempted.

We obtained ethical approval for this study from the Ethical Committee of the Origyn Fertility Center, the agreement being signed by the Medical Director of the centre as well. Confidentiality was granted through and after the study. Written informed consent was obtained from the couple for the publication of this case report and any accompanying images.

## 3. Discussion

Here we report the case of a Romanian couple, both of them being carriers for the congenital disorder of glycosylation type Ia (CDG-Ia). The first foetus died two months after birth. Imperative for CDG-Ia, the second baby has died at six months, the necropsy exam revealing a hyperaccumulation of fluid in the cerebellum, as well as a slow blood flow in the kidneys, lungs, and liver, respectively, and an increased amount of blood in the cerebellum, brain, liver, and lung. Additionally, abnormalities, such as the persistence of the arterial canal and ventricular septal defect have been observed.

CDG-Ia is an inborn condition which usually affects every system of organs, its type and magnitude ranging between individuals, even within the same family [31]. Depending on the mutation, there have been identified distinct phenotypes, varying from physical dysmorphism and intellectual disability, to polycystic kidney syndrome and thrombosis [32].

Vega et al. [10] aimed to reveal the molecular pathogenesis using a prokaryotic expression system. Therefore, from a small group consisting of 22 Spanish PMM2 patients, and the analysis of fourteen nucleotide change, they discovered six loss-of-function proteins, seven having residual activities up to more than 50%, and one normal variant change.

In 9 out of 54 patients, Grünewald et al. [33] observed similar residual activities of proteins in fibroblasts up to 70%. An abnormal transferrin pattern and a high reduced PMM activity have defined this subgroup. However, a subset of six patients were characterised by reduced PMM activity, which proved to be the result of the mutation that occurred in C241S.

Based on study design and methodology, the results obtained by Citro et al. [34] are somehow in an antithetical phase with the previous two studies. They have demonstrated that PMM2 is very tolerable to mutations and that the CDG-related phenotype could be the result of mutations occurring in other associated genes(s). One potential inhibitor for missense mutations is CID2876053 (1-(3-chlorophenyl)-3,3-bis(pyridine-2-yl)urea). It has a strong binding affinity for D65Y mutant but also displayed a strong interaction with the I132N, I132T, and 183S [35].

The biochemical parameters tested were in normal limits, according to the results obtained by Casado et al. [36]. However, the cranial magnetic resonance imaging (MRI) revealed distinct particularities in both patients. Both babies were heterozygous, but they had different features regarding the transferrin pattern following the tests performed. The suspicion of PMM2-CDG had been confirmed based on two atypical features such as tremor and the global cerebellar atrophy with vermis hypoplasia.

Considering that CDG-Ia affects several systems of organs, around 20% of infants usually do not survive the first months [37]. Unfortunately, the parents that are carriers for certain mutation(s) often find out when they conceive a child. In our case, the couple found out after the death of their second child.

Analogous to our case, we have started from the presumption that, according to which, each unexplained HF could represent a relevant pointer when we talk about PMM2-deficiency. Thus, we have found it appropriate to summarize the existing literature regarding the accumulation of fluid within the body compartments, as shown in Table 1. Moreover, it must be taking into consideration that mutations of ALG1 [38,39], ALG8 [40], Ih [40], and Ix [26], have been also associated with HF, but also with ascites or edema.

On the basis of Table 1, HF was reported in three studies [41,43,44], but it should also be mentioned that skin edema is usually considered as the first clinical sign specific to CDG [42,45]. Along with the previously mentioned clinical signs, pericardial effusion, large heart, thrombocytopenia, and ascites have also been encountered in cases of cardiomegaly [41,42,43,44,45].

Schollen et al. [46] have demonstrated that, from 92 independent pregnancies, in which both partners were at risk for CDG-Ia, the percentage of affected foetuses has been around 34%. This could be explained by the high frequency of the R141H mutation in the PMM2 gene, concluding that the recurrence risk is close to one-in-three in CDG-Ia families.

Unlike the westernised countries in which such genetic investigations are reimbursed through the public health system [47], for a mid-class country, such as Romania, the costs are quite prohibitive. On the other hand, the field of assisted human reproduction has expanded significantly in Romania in recent years, an IVF national program being developed in this context [48].

Based on the established methodology and following the transfer of the frozen embryo(s), only a biochemical pregnancy was obtained, despite our best efforts. Of the six embryos tested, three were euploids, one of which had compound heterozygosity (PMM2 at exon 5 and 6) and the other two were simple heterozygotes (PMM2 at exon 5 and 6).

Despite the fact that preimplantation genetic testing (PGT) is commonly used in current clinical practice, this is the first case of IVF associated with PGT-M, according to our best of knowledge.

## 4. Conclusions

It can be concluded, based on the results of our case report and of the existing literature, that CDG-Ia is one of the most debilitating inherited disorders, which affects the entire homeostasis of the organism. Although no therapy or treatment has been developed, we consider it suitable to report the outcome of a PGT-M associated with IVF. This may increase awareness regarding the importance of developing a sustainable strategy for rare disorders such as CDG-Ia in the near future.

## Figures and Tables

**Table 1 genes-11-00697-t001:** The outcomes and consequences related to PMM2 gene mutation(s).

PMM2-CDG Phenotype	Pregnancy Status	Molecular Analysis	Reference
Postnatal	Full-term (death at four weeks) and ultrasound echography at twenty gestational weeks	c.691G>A p.Val231Met c.710C>G p.Thr237Arg	[41]
At fifty days	Full-term (death at eight weeks) and ultrasound echography at birth	c.357CNA p.Phe119Leu c.470 TNC p.Phe157Ser	[42]
Both in day two	Births at thirty-two and thirty-six gestational weeks, respectively, through Caesarean section. Death at day 7 and an ultrasound echography was carried out at twenty-nine gestational weeks. Death in week eight and an ultrasound echography was carried out at thirty-five gestational weeks	c.160_161insG c.357CNA p.Phe119Leu c.357CNA p.Phe119Leu c.470 TNC p.Phe157Ser	[43]
After birth, before week eight	Siblings—Both births at thirty-six gestational weeks through Caesarean section and have died at three and eight weeks, respectively. Only one ultrasound echography was carried out at thirty-one gestational weeks. Not specified for the other one.	c.161_162insG c.385GNT p.Val129L	[44]
Not specified	Full-term from which one died at six years and the third at four months. Not specified for the second infant. An ultrasound echography was carried out at birth for all three	c.357CNA p.Phe119Leu c.422GNA p.Arg141His c.691GNA p.Val231Met c.640-15479CNT c.563ANG p.Asp188Gly c.104 T>A p.Leu35	[45]

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
