# Peer review of "In Vitro Fertilisation (IVF) Associated with Preimplantation Genetic Testing for Monogenic Diseases (PGT-M) in a Romanian Carrier Couple for Congenital Disorder of Glycosylation Type Ia (CDG-Ia): A Case Report"

_genes, 2020, doi:10.3390/genes11060697_

Round 1
Reviewer 1 Report
This case report presents the use of preimplantation genetic testing after In Vitro Fertilisation in a carrier couple for Congenital Disorder of Glycosylation Type Ia.
There are no previous similar cases reported in literature, although the use of preimplantation testing for monogenic diseases (PGT-M) is actually widespread and of common practice.
The title properly describe the core message of the case, still it should be completed with the phrase “A case report”.
There are no concerns about ethical aspect.
The abstract is adeguatelly written, summarizing the key message with necessary detail in a concise manner; it contains a misprint “Here we report we outcome “ (line 29) and should be corrected.
The introduction well indicates the context of the report and it gives sufficient information about the disease of interest. Yet it should be here emphasized the novelty of the case.
The case description results appropriate in providing details of the history and the procedures. There are some questions about missing data: line 133-135 and 148
“A total of 11 oocytes had normal fertilised, and by day 5, 3 of the embryos had degenerated. Furthermore, 7 of the embryos were submitted to biopsy, 3 of them on day 5 and the other 4 on day 6 at blastocysts stage. [..] Of the 6 embryos analysed by PGT-M”
You don’t mention why only 7 of 8 normal fertilised embryos were submitted to biopsy, maybe one more embryo degenerated before the blastocyst stage and you didn’t specify it? Furthermore you indicate 7 biopsies performed, but later you describe only 6 embryos analysed by PGT-M, what happened with the seventh embryo biopsied?
The intent of Table 1 is to describe the outcomes and consequences related to PMM2 gene mutation(s), related to the specification in the text “summarize the existing literature regarding the accumulation of fluid within body compartments”. Subsequently you would expect a description of the clinical findings, while in the table is only reported if an ultrasound echography or more were performed. It should be hypotized that all these ultrasound scans revealed accumulation of fluid within body compartments? Better explanation is needed.
In conclusion the case report is satisfactorily written, with some ameliorating specifications needed.
Author Response
Dear Reviewer #1,
Thank you very much for the positive feedback, interest, and time spent reviewing our manuscript. We appreciate the valuable comments which have helped us to create a much more appropriate case report and, therefore, to provide a clearer perspective.
Here is a list of all the changes we made:
1. The title properly describe the core message of the case, still it should be completed with the phrase “A case report”.
In Vitro Fertilisation (IVF) Associated with Preimplantation Genetic Testing for Monogenic Diseases (PGT-M) in a Romanian Carrier Couple for Congenital Disorder of Glycosylation Type Ia (CDG-Ia): A case report
2. The abstract is adeguatelly written, summarizing the key message with necessary detail in a concise manner; it contains a misprint “Here we report we outcome “ (line 29) and should be corrected.
Here we report the outcome of an in vitro fertilisation (IVF) cycle associated with preimplantation genetic testing for monogenic diseases (PGT-M) in a Romanian carrier couple for CDG type Ia with distinct mutations of the PMM2 gene.
3. The introduction well indicates the context of the report and it gives sufficient information about the disease of interest. Yet it should be here emphasized the novelty of the case.
Despite the fact that preimplantation genetic testing (PGT) is commonly used in the current clinical practice, this is the first case of IVF associated with PGT-M according to our best of knowledge.
4. The case description results appropriate in providing details of the history and the procedures. There are some questions about missing data: line 133-135 and 148
- “A total of 11 oocytes had normal fertilised, and by day 5, 3 of the embryos had degenerated. Furthermore, 7 of the embryos were submitted to biopsy, 3 of them on day 5 and the other 4 on day 6 at blastocysts stage. [..] Of the 6 embryos analysed by PGT-M”
- You don’t mention why only 7 of 8 normal fertilised embryos were submitted to biopsy, maybe one more embryo degenerated before the blastocyst stage and you didn’t specify it? Furthermore you indicate 7 biopsies performed, but later you describe only 6 embryos analysed by PGT-M, what happened with the seventh embryo biopsied?
A total of 11 oocytes had normal fertilised, and by day 5, 4 embryos degenerated. Therefore, 7 embryos have been biopsied, 3 of them on day 5 and the other 4 on day 6 at blastocysts stage.
Of the 7 embryos analysed by PGT-M, only 3 embryos were euploids, one of which presented compound heterozygosity (PMM2 exon 5 and PMM2 exon 6), while the other two were simple heterozygotes, one PMM2 exon 5, and the other PMM2 exon 6.
(We are sorry. There were 7 embryos. It was a typing error in both cases. Thank you.)
5. The intent of Table 1 is to describe the outcomes and consequences related to PMM2 gene mutation(s), related to the specification in the text “summarize the existing literature regarding the accumulation of fluid within body compartments”. Subsequently you would expect a description of the clinical findings, while in the table is only reported if an ultrasound echography or more were performed. It should be hypotized that all these ultrasound scans revealed accumulation of fluid within body compartments? Better explanation is needed.
Analogous to our case, we have started from the presumption according to which each unexplained HF could represent a relevant pointer when we talk about PMM2-deficiency. Thus, we have found it appropriate to summarize the existing literature regarding the accumulation of fluid within the body compartments (Table 1). On the basis of Table 1, HF was reported in 3 studies [41,43,44], but it should also be mentioned that skin edema is usually considered as the first clinical sign specific to CDG [42,45]. Along with the previously mentioned clinical signs, also have been encountered cardiomegaly with pericardial effusion, large heart, thrombocytopenia, and ascites [41–45].
We would like to thank you again very much.
Kind regards,
Ovidiu-Dumitru Ilie
Reviewer 2 Report
The manuscript entitled “In Vitro Fertilisation (IVF) Associated with Preimplantation Genetic Testing for Monogenic Diseases (PGT-M) in a Romanian Carrier Couple for Congenital Disorder of Glycosylation Type Ia (CDG-Ia)” by Doroftei and coworkers, is a case report that describes the outcome of an in vitro fertilisation cycle associated with preimplantation genetic testing in a Romanian carrier couple, both carriers of CDG type Ia, the most common type among the CDGs.
PMM2-CDG is caused by mutations in PMM2 and more than 110 different pathological mutations have been so far identified. The missense mutations may affect the structure and/or the enzymatic activity of the protein, some mutations are more severe than others. Moreover moreover the effect of other genes, and in particular of those involved in the pathway of N-glycosylation, that can act as “modifiers” should also be considered.
The case herein described, concerns with two healthy carriers bearing the mutations V129M and F157S. Both these mutations are classified as pathogenic although mild clinical profiles are typically reported.
Since the outcome of the in vitro fertilisation could be influenced by the specific mutation of PMM2 and by the overall genetic context, I would suggest commenting on both these aspects.
Hopefully, the molecular and genetic features could be helpful in the future for successful applications of in vitro fertilisation combined with preimplantation genetic testing (for example for the selection of the most suitable embryo to transfer).
Here there are few references that could be helpful for this discussion:
https://pubmed.ncbi.nlm.nih.gov/22012410/
https://www.tandfonline.com/doi/abs/10.1080/07391102.2019.1708797?needAccess=true&journalCode=tbsd20
https://pubmed.ncbi.nlm.nih.gov/30061496/
https://pubmed.ncbi.nlm.nih.gov/22012410/
https://pubmed.ncbi.nlm.nih.gov/21541725/
Minor points:
Few typing mistakes:
Line 29 ... we report we outcome…
Line 144 … or Cy3-Dutp…
Author Response
Dear Reviewer #2,
Thank you very much for the positive feedback, interest, and time spent reviewing our manuscript. We appreciate the valuable comments which have helped us to create a much more appropriate case report and, therefore, to provide a clearer perspective.
Here is a list of all the suggested references:
Vega et al., [10] (https://pubmed.ncbi.nlm.nih.gov/21541725/) aimed to reveal the molecular pathogenesis using a prokaryotic expression system. Therefore, from a small group consisting of 22 Spanish PMM2 patients and the analysis of fourteen nucleotide change, they have discovered six loss-of-function proteins, seven having residual activities up to more than 50% and one normal variant change.
In 9 out of 54 patients, Grünewald et al., [33] (https://core.ac.uk/download/pdf/81166959.pdf) observed similar residual activities of proteins in fibroblasts up to 70%. An abnormal transferrin pattern and a high reduced PMM activity have defined this subgroup. However, a subset of 6 patients has been characterised by a reduced PMM activity, which proved to be the result of the mutation that occurred in C241S.
Based on study design and methodology, the results obtained by Citro et al., [34] (https://pubmed.ncbi.nlm.nih.gov/30061496/) are somehow in an antithetical phase with the previous two studies. They have demonstrated that PMM2 is very tolerable to mutations, and the CDG-related phenotype could be the result of mutations occurred in other associated genes (s). One potential inhibitor for missense mutations is CID2876053 (1-(3-chlorophenyl)-3,3-bis(pyridine-2-yl)urea). It has a strong binding affinity for D65Y mutant but also displayed a strong interaction with the I132N, I132T, and 183S [35] (https://www.tandfonline.com/doi/abs/10.1080/07391102.2019.1708797?needAccess=true&journalCode=tbsd20)
The biochemical parameters tested were in normal limits according to the results obtained by Casado et al [36] (https://pubmed.ncbi.nlm.nih.gov/22012410/). However, the cranial magnetic resonance imaging (MRI) revealed distinct particularities in both patients. Both babies were heterozygous, but they had different features regarding the transferrin pattern following the tests performed. Considering that tremor and the global cerebellar atrophy with vermis hypoplasia were two atypical characteristics, in this context has been strengthened the suspicion of PMM2-CDG disease.
Line 29 ... we report we outcome…
Here we report the outcome of an in vitro fertilisation (IVF) cycle associated with preimplantation genetic testing for monogenic diseases (PGT-M) in a Romanian carrier couple for CDG type Ia with distinct mutations of the PMM2 gene.
Line 144 … or Cy3-Dutp…
Cy3-dUTP.
We would like to thank you again very much.
Kind regards,
Ovidiu-Dumitru Ilie